Random allogeneic blood transfusion in pigs: characterisation of a novel experimental model

Ziebart Alexander alexander.ziebart@unimedizin-mainz.de 1
Schaefer Moritz M. 1
Thomas Rainer 1
Kamuf Jens 1
Garcia-Bardon Andreas 1
Möllmann Christian 1
Ruemmler Robert 1
Heid Florian 1
Schad Arno 2
Hartmann Erik K. 1
1 Department of Anesthesiology, Medical Centre of the Johannes Gutenberg-University , Mainz , Germany
2 Institute of Pathology, Medical Centre of the Johannes Gutenberg-University , Mainz , Germany
Kim Cheorl-Ho
Electronic publication date: 2019 Aug 16
Publication date: 2019
Volume: 7
Electronic Location ID: e7439
Received 2018 Oct 18; Accepted 2019 Jul 8
Copyright: ©2019 Ziebart et al.
Copyright year: 2019
Copyright holder: Ziebart et al.
License: This is an open access article distributed under the terms of the Creative Commons Attribution License, which permits unrestricted use, distribution, reproduction and adaptation in any medium and for any purpose provided that it is properly attributed. For attribution, the original author(s), title, publication source (PeerJ) and either DOI or URL of the article must be cited.
License URL: https://creativecommons.org/licenses/by/4.0/

Keywords: Transfusion, Blood, Pig modell, Crosstalk, ARDS, Lung, Brain, Inflammation, Experimental model

Funding: Mainz Research School of Translational Biomedicine (TransMed) Johannes Gutenberg University, Mainz, Germany The study was funded by a fellowship of Mainz Research School of Translational Biomedicine (TransMed) of the Johannes Gutenberg University, Mainz, Germany. The funders had no role in study design, data collection and analysis, decision to publish, or preparation of the manuscript.

==============================
Background

Organ cross-talk describes interactions between a primary affected organ and a secondarily injured remote organ, particularly in lung-brain interactions. A common theory is the systemic distribution of inflammatory mediators that are released by the affected organ and transferred through the bloodstream. The present study characterises the baseline immunogenic effects of a novel experimental model of random allogeneic blood transfusion in pigs designed to analyse the role of the bloodstream in organ cross-talk.

Methods

After approval of the State and Institutional Animal Care Committee, 20 anesthetized pig were randomized in a donor and an acceptor (each n = 8): the acceptor animals each received high-volume whole blood transfusion from the donor (35–40 ml kg−1). Four animals received balanced electrolyte solution instead of blood transfusion (control group; n = 4). Afterwards the animals underwent extended cardiorespiratory monitoring for eight hours. Post mortem assessment included pulmonary, cerebral and systemic mediators of early inflammatory response (IL-6, TNF-alpha, iNOS), wet to dry ratio, and lung histology.

Results

No adverse events or incompatibilities occurred during the blood transfusion procedures. Systemic cytokine levels and pulmonary function were unaffected. Lung histopathology scoring did not display relevant intergroup differences. Neither within the lung nor within the brain an up-regulation of inflammatory mediators was detected. High volume random allogeneic blood transfusion in pigs neither impaired pulmonary integrity nor induced systemic, lung, or brain inflammatory response.

Conclusion

This approach can represent a novel experimental model to characterize the blood-bound transmission in remote organ injury.

Introduction

Single organ injury or failure does not exert an isolated effect on the primary affected organ system. Remote organs consecutively develop secondary injuries that are not connected to the initial insult. This is referred to as deleterious organ cross-talk and described particularly for lung and brain interactions, where this pathomechanism is well known, but not fully understood. For example, mechanical ventilation in the intensive care unit may cause inflammatory response and induce damage to distant organs. On the other hand, severe brain injury is associated with an increased risk of lung failure (Mascia et al., 2007; Pelosi & Rocco, 2011).

Different theories for this phenomenon consider neuronal response, hormone release or circulating inflammatory mediators as causative (Han, 2017; Mrozek, Constantin & Geeraerts, 2015; Pelosi & Rocco, 2011). The hypothesis of inflammatory mediator distribution is promising as several studies show inflammatory response after a primary hit. The blood stream seems to be the distributor in remote organ injury and cross-talk (Hegeman et al., 2009; Klein et al., 2016; Mrozek, Constantin & Geeraerts, 2015). In theory, inflammatory cytokines that are released into the systemic circulation induce remote organ injury. Hence, for experimental purposes this effect might be transferable from one individual to another via whole blood transfusion. Porcine models, in this context, offer advantages in experimental assessment of blood stream-related distribution of inflammatory mediator transmission. Firstly, higher blood volumes can be transferred in comparison to common laboratory rodent species. Secondly, because major inter-individual blood incompatibilities are missing (Smith et al., 2006). Only few reports reveal bronchospasm, disseminated intravascular coagulation, bleeding, or progressive hypotension during blood transfusion in pigs (Hunfeld et al., 1984; Sheil et al., 1972; Smith et al., 2006). This porcine system is based on blood groups A0, whereas the human group B is missing. Additional 16 different antigens are known to form the subgroups A-P (Smith et al., 2006). Unlike the human Rhesus antigen, which is highly immunogenic and prone for transfusion reactions or hemolysis, the porcine Rhesus antigen is coded by a single gene without any polymorphisms and less pathologic relevance (Avent & Reid, 2000; Omi et al., 2003). However, the distinct effects of porcine blood transfusion with focus on end-organ inflammatory response are barely examined. In order to further elucidate the role of the blood stream as a distributor in remote organ injury the present study aims to characterize the baseline immunogenic effects of random allogeneic blood transfusion in pigs. The approach may allow for experimental separation of blood-bound effects on organ integrity from an already injured subject: hence, we hypothesized that single, large-volume blood transfusion from a healthy donor neither exhibits inflammatory response nor impairs the pulmonary function in the acceptor. As secondary end points, the systemic and cerebral inflammatory response is analyzed.

Materials & Methods

The prospective-randomized animal study was approved by the State and Institutional Animal Care Committee (Landesuntersuchungsamt Rheinland-Pfalz, Koblenz, Germany; approval number: 23 177-07∕G 14-1-084). The study was conducted in accordance with the ARRIVE guidelines and the international guidelines for the care and use of laboratory animals. The experiments were performed from 12/2014 till 3/2015. Twenty juvenile German Landrace pigs (sus scrofa domestica, weight 28 ± 2 kg) were examined. Eight animals served as blood donors. Eight animals received allogeneic blood transfusion. The control group consisted of four animals that received the same amount of balanced electrolyte solution instead of blood transfusion. One acceptor animal was excluded due to complicated airway management and multiple intubation attempts with subsequent respiratory failure before the transfusion. Per protocol and approval, the donor animals that inevitably suffered from hemorrhagic shock were subsequently included in another study protocol focusing on fluid resuscitation. All experiments were conducted under continuous general anesthesia. This manuscript adheres to the applicable EQUATOR guidelines.

Anesthesia and instrumentation

Anesthesia and instrumentation procedure was previously described in detail in our former studies (Ziebart et al., 2015; Ziebart et al., 2018): Briefly, in each case two animals received an intramuscular injection of ketamine (8 mg kg−1) and midazolam (0.2 mg kg−1) followed by induction of general anesthesia via an ear vein cannula (propofol 4 mg kg−1, and fentanyl 4 µg kg−1, atracurium 0.5 mg kg−1), which was maintained by continuous infusion (propofol 8–12 mg kg−1 h−1, fentanyl 0.1–0.2 mg h−1). After orotracheal intubation the animals were ventilated in volume-controlled mode (VCV; AVEA, CareFusion, USA): positive end-expiratory pressure (PEEP) five cmH2O, tidal volume 7 ml kg−1, inspiration to expiration ratio 1:2, fraction of inspired oxygen (FiO2) 0.4 and variable respiratory rate to achieve an end-tidal CO2 <45 mmHg. In case of respiratory insufficiency, mechanical ventilation parameters were escalated according to an adaption of the ARDS network protocol (Ziebart et al., 2014). Afterwards, the animals were randomized into an acceptor and a donor animal. Ultrasound guided femoral vascular access under sterile conditions included a central venous line, a pulse contour cardiac output catheter (PiCCO, Pulsion Medical, Germany), and a pulmonary arterial catheter in the acceptor animals. The donor animals received a central venous catheter, an arterial line, a pulse contour cardiac output system (PiCCO, Pulsion Medical Systems, Germany) and a large-bore venous introducer. Extended hemodynamics, transpulmonary thermodilution-derived parameters, spirometry and gas exchange were continuously monitored (Datex S/5 Monitor, GE Healthcare, Germany; PiCCO, Pulsion Medical Systems, Germany). Hematological and blood gas measurements were repetitively conducted. Electrical impedance tomography (EIT; Goe-MF II, CareFusion, Germany) was used to measure variations of the thoracic bioimpedance that are associated with pulmonary aeration. Sixteen adhesive electrodes were placed around the thorax and the ventilation distribution was measured for three lung compartments (Level 1-3; non-dependent, central, dependent) (Bodenstein et al., 2012; Ziebart et al., 2014). Multiple breath-nitrogen washout/washin technique was applied to measure the functional residual capacity (Kamuf et al., 2017). Body temperature was measured by a rectal probe and a surface warming device prevented hypothermia. All animals received a background infusion of 5 ml kg−1 h−1 balanced electrolyte solution (Sterofundin; B.Braun, Melsungen, Germany).

After anesthesia, instrumentation, and 20-minute recovery period baseline values were measured and recorded.

Blood withdrawal procedure

From the donor animals 35–40 ml kg−1 blood was taken for allogeneic blood transfusion through the arterial catheter. The whole blood was collected in specific collector bags system (Composelect, Fresenius-Kabi AG, Homburg Germany). These bags comprised 63 ml CPD/100 ml SAG-M - RCC (Citrate phosphate dextrose, Erythrocyte storage in hypertonic conservation medium). After collection in a primary bag, the blood was drained through a leucocyte-depleting filter into a second bag that contained the anticoagulants. Finally, the whole blood was collected into a bag with a connector for a transfusion system. Afterwards, the whole blood was stored by room temperature for about ten minutes before transfusion to avoid negative effects of long-time storing or cooling. Afterwards the donor animals were transferred to the fluid resuscitation protocol that was included in the same approval: they were euthanized by intravenous injection of propofol (200 mg) and potassium chloride (40 mval) within the scope of this protocol (Ziebart et al., 2018).

Experimental protocol

The experimental protocol is summarized in Fig. 1. Immediately after blood withdrawal the whole blood was transfused into the recipients through the venous vascular access sheath over 15 min through a transfusion system with an integrated aggregate-filter. A control group (n = 4) underwent the identical instrumentation and protocol, but received an equal amount of balanced electrolyte solution instead of allogeneic blood transfusion. Afterwards, the animals were monitored for eight hours. In case of hemodynamic deterioration (mean arterial pressure <60 mmHg), continuous noradrenaline infusion was administered. After finishing the protocol, the animals were killed in deep general anesthesia by injection of propofol (200 mg) and potassium chloride (40 mval).

Figure 1 Experimental flow chart.

After preparation and baseline measurement the animals were observed for eight hours. The hemodynamic and spirometric parameters were recorded continuously. Blood samples were taken every two hours. EIT (electrical impedance tomography) was measured hourly. Histopathological analysis of the lung was performed after the end of the experiment. Pulmonary and cerebral mRNA expression was quantified (Transfusion n = 7; Control n = 4).

Post mortem analysis

Repetitively taken serum blood samples were used to determine the serum levels of TNF-alpha and IL-6 by enzyme-linked immunosorbent assays (Porcine Quantikine ELISA Kits; R/D Systems, Wiesbaden, Germany). After finishing the protocol, the lung and the brain each were removed en-bloc. Four representative tissue samples (dependent periphery, dependent central, non-dependent central, non-dependent periphery) from the right lung were extracted, formalin-fixed, and used for histological evaluation of alveolar damage in the upper and the lower lung. After paraffin embedding and hematoxylin/eosin staining the samples were analyzed in investigator-blinded manner by means of standardized scoring scheme (Ziebart et al., 2015; Ziebart et al., 2014). Additional lung samples were cryopreserved for mRNA expression analysis of inflammatory mediators (IL-6, TNF-alpha, iNOS) using real-time polymerase chain reaction (rt-PCR; Lightcycler 480 PCR System; Roche Applied Science, Penzberg, Germany) as described in detail by Hartmann et al. (2015). The mRNA expression was normalized to peptidylprolyl isomerase A (PPIA). The same technique and markers were used to quantify the cerebral mRNA expression from the right hippocampus and the right frontal cortex. Furthermore, the lung and brain water content were determined through the tissue wet to dry ratio. For this procedure the weight of the removed organs was determined directly and after two days of complete drying.

Statistics

All parameters are presented as mean and standard deviation (± SD) or displayed as plots. The analysis focusses on the relevant time points baseline, 0 h, 4 h 8 h. Additional data are added as supplemental files. Group effects over time (Transfusion, Control) were compared by two-way analysis of variance (ANOVA) with post-hoc Student-Newman-Keuls-Test. Mann–Whitney-U-Test was used to analyze the post mortem parameters. A p-value lower than 0.05 was accepted as significant. The software package SigmaPlot 12.5 (Systat Software, San Jose, CA, USA) was used.

Results

The estimated total blood volume of the pigs (74 ml kg−1) was 2084 ± 166 ml (Lyhs & Wachtel, 1966). The transfused blood volume was 48 ± 4%. No adverse events or major incompatibilities occurred during the blood transfusion procedures. Apart from a temporary increase of the mean pulmonary arterial pressure following blood transfusion and a slight increase of the central venous pressure the cardiopulmonary monitoring and blood gas analyses showed no relevant differences between the transfusion and control animals over eight hours (Tables 1–3). The lactate level was in a normal range and even decreased within eight hours. Following the blood transfusion hemoglobin, hematocrit and erythrocyte counts significantly increased without relevant changes in leucocyte amounts (Fig. 2). Markers of systemic inflammatory response did not differ between the groups as well (Figs. 2 and 3). The pulmonary function as measured by blood gas analysis and ventilation distribution was not affected by blood transfusion (Table 3 & Fig. 4). Only the alveolar-arterial oxygen difference increased. At baseline, two animals started at a PaO2/FiO2 <400 mmHg without signs of local or systemic illness. A modification of the initial respiratory setting due to hypoxemia was not required in any animal. Lung histopathology scoring did not display relevant intergroup differences (control vs. transfusion; 10 ± 1 vs. 11 ± 3; each p > 0.05). The tissue wet to dry ratio as surrogate of edema formation between both groups was not increased following transfusion (control vs. transfusion; lung: 4.7 ± 0.2 vs. 4.6 ± 0.2; brain: 3.3 ± 0.3 vs. 3.4 ± 0.3; each p > 0.05). Neither within the lung nor within the brain the mRNA expression analysis revealed an up-regulation of inflammatory markers (Figs. 5 and 6).

Table 1 Hemodynamic parameters.

No significant intra- and intergroup changes of hemodynamic parameters were observed after blood transfusion or fluid infusion, with the exception of a temporary pulmonary arterial pressure increase and raise of the central venous and mean arterial pressure after transfusion/fluid administartion. (Transfusion n = 7; Control n = 4).

Parameter		BLH	0 h	4 h	8 h	
	 	MEAN (SD)	MEAN (SD)	MEAN (SD)	MEAN (SD)	
MAP	Transfusion	70 (13)	98 (14)*	82 (9)	81 (14)	
[mmHg]	Control	77 (12)	93 (3)*	73 (7)	71 (5)	
HR	Transfusion	83 (19)	83 (13)	84 (20)	85 (13)	
[min−1]	Control	78 (9)	78 (10)	80 (7)	73 (6)	
PAP	Transfusion	18 (5)	27 (6)*,#	20 (2)	19 (3)	
[mmHg]	Control	16 (4)	19 (3)	17 (3)	12 (7)	
Pes	Transfusion	5 (1)	7 (2)	4 (1)	3 (2)	
[mmHg]	Control	6 (0)	6 (1)	4 (1)	4 (0)	
CO	Transfusion	3.6 (0.7)	3.7 (0.7)	3.2 (0.3)	3.2 (0.6)	
[l min−1]	Control	3.4 (0.3)	4.3 (0.9)	3.4 (0.3)	2.9 (0.4)	
GEDI	Transfusion	820 (125)	850 (96)	778 (107)	812 (81)	
[ml kg−1]	Control	820 (97)	916 (20)	808 (62)	803 (75)	
EVLWI	Transfusion	11 (2)	12 (2)	12 (2)	12 (3)	
[ml kg−1]	Control	13 (5)	11 (0)	15 (3)	13 (2)	
CVP	Transfusion	5 (2)	9 (2)*	7 (2)*	7 (2)*	
[mmHg]	Control	7 (1)	8 (2)	7 (1)	7 (1)	
SpO2	Transfusion	100 (0)	100 (0)	99 (1)	99 (1)	
[%]	Control	100 (0)	100 (0)	100 (0)	100 (0)	
Notes.

* indicates p < 0.05 vs. baseline value.

# indicates p < 0.05 in intergroup comparison.

MAP mean arterial pressure

HR heart rate

PAP mean arterial pulmonary pressure

Pes esophageal pressure

CO cardiac output

GEDI global enddiastolic volume index

EVLWI extravascular lung water index

CVP central venous pressure

SpO2 oxygen saturation

Table 2 Spirometric parameters.

No significant intra- and intergroup changes of spirometric parameters were observed after blood transfusion or fluid infusion (Transfusion n = 7; Control n = 4).

Parameter	 	BLH	0 h	4 h	8 h	
	 	MEAN (SD)	MEAN (SD)	MEAN (SD)	MEAN (SD)	
FiO2	Transfusion	39 (0)	39 (0)	39 (0)	39 (0)	
[%]	Control	39 (0)	39 (0)	39 (0)	39 (0)	
MV	Transfusion	6.2 (0.7)	6.1 (0.8)	6.2 (1)	6.6 (1.3)	
[l/min]	Control	6.4 (1.7)	5.8 (1.3)	6.5 (1.6)	7 (1)	
TV	Transfusion	8 (0)	8 (0)	8 (0)	8 (0)	
[ml kg−1]	Control	8 (1)	8 (1)	8 (1)	8 (1)	
Ppeak	Transfusion	17 (2)	18 (2)	20 (2)	20 (2)	
[cm H2O]	Control	16 (2)	17 (1)	17 (3)	19 (3)	
Pmean	Transfusion	9 (1)	9 (0)	10 (1)	10 (1)	
[cm H2O]	Control	9 (1)	8 (0)	9 (1)	10 (1)	
PEEP	Transfusion	4 (0)	4 (0)	4 (0)	4 (0)	
[cm H2O]	Control	4 (0)	4 (0)	4 (0)	4 (0)	
Cp	Transfusion	19 (3)	17 (2)	16 (2)	16 (3)	
[ml/cmH2O]	Control	20 (3)	20 (2)	19 (3)	17 (3)	
RAW	Transfusion	10 (1)	11 (1)	12 (1)	12 (1)	
[kPA/l/sec]	Control	11 (1)	11 (1)	11 (1)	11 (2)	
FRC	Transfusion	728 (206)	691 (260)	691 (240)	714 (235)	
[ml]	Control	823 (369)	610 (87)	674 (77)	788 (351)	
Notes.

* indicates p < 0.05 vs. baseline value.

# indicates p < 0.05 in intergroup comparison.

FiO2 fraction of inspired oxygen

MV minute volume

TV tidal volume

Ppeak peak inspiratory pressure

Pmean mean airway pressure

PEEP positive end-expiratory pressure Cp: pulmonary compliance

RAW airway resistance; FRC: functional residual capacity

Table 3 Blood gas parameter.

No relevant intra- and intergroup changes of blood gas parameters were observed after blood transfusion or fluid infusion. Only the lactate levels decrease within the normsl range and the AaDO 2 increases in both groups (Transfusion n = 7; Control n = 4).

Parameter	 	BLH	0 h	4 h	8 h	
	 	MEAN (SD)	MEAN (SD)	MEAN (SD)	MEAN (SD)	
SvO2	Transfusion	60 (12)	66 (8)	58 (6)	56 (6)	
[%]	Control	63 (13)	64 (0)	63 (11)	53 (8)	
pH	Transfusion	7.47 (0.028)	7.505 (0.027)	7.503 (0.017)	7.495 (0.019)	
 	Control	7.485 (0.02)	7.5 (0.033)	7.545 (0.054)	7.506 (0.069)	
BE	Transfusion	4.5 (4)	7.3 (1.4)	8.5 (2.2)	6.6 (3.6)	
[mmol/ml]	Control	5.9 (4.5)	6.1 (0)	8.9 (2.4)	6.4 (3.4)	
artCO2	Transfusion	40 (4)	40 (4)	41 (3)	40 (5)	
[mmHg]	Control	39 (5)	41 (4)	38 (6)	38 (4)	
etCO2	Transfusion	37 (3)	38 (3)	40 (1)	39 (3)	
[mmHg]	Control	35 (5)	35 (4)	34 (5)	35 (4)	
avCO2 diff	Transfusion	9 (3)	8 (4)	8 (2)	5 (6)	
[mmHg]	Control	10 (5)	−8 (21)	7 (1)	8 (1)	
PaO2	Transfusion	185 (34)	181 (23)	154 (33)	141 (26)	
[mmHg]	Control	206 (43)	230 (49)	186 (44)	171 (24)	
PaO2/FiO2	Transfusion	464 (86)	452 (58)	386 (81)	352 (64)	
[mmHg]	Control	514 (108)	576 (122)	464 (109)	428 (60)	
AaDO2	Transfusion	53 (30)	58 (22)	82 (32)*	97 (27)*	
[mmHg]	Control	33 (44)	7 (46)	55 (45)	69 (27)*	
Temp.	Transfusion	37.1 (0.8)	36.9 (1.2)	38 (0.8)	38.5 (0.4)	
[∘C]	Control	35.8 (2.9)	35.4 (3.9)	37.6 (2.2)	38.3 (0.6)	
Potassium	Transfusion	4 (0.3)	3.6 (0.7)	4.2 (0.7)	3.9 (0.6)	
[mmol/l]	Control	4 (0.6)	4 (0.1)	4.5 (0.3)	4.2 (0.2)	
Lactate	Transfusion	1.4 (0.5)	1.1 (0.4)	0.7 (0.2)*	0.6 (0.2)*	
[mmol/l]	Control	1.4 (0.4)	1.3 (0.1)	0.9 (0.2)*	0.8 (0.3)*	
Notes.

* indicates p < 0.05 vs. baseline value.

# indicates p < 0.05 in intergroup comparison.

SvO2 central venous oxygen saturation

BE base excess

art.CO2 arterial carbon dioxide

ex.CO2 exspiratory carbon dioxide

avCO2 diff arterial venous carbon dioxide difference

PaO2 arterial oxygen

PaO2/FiO2 Horovitz index

AaDO2 alveolar-arterial oxygen difference

Temp. Temperature

Figure 2 Hematological parameters. (A, leucocytes; B, hemoglobin, C, erythrocytes; D, hematocrit; E, thrombocytes; F, neutrophil granulocytes).

Parameters are displayed as percentage of baseline. A significant increase of hemoglobin, hematocrit and erythrocytes was observed after transfusion. The decrease of these parameters in the control group was not significant; *=p < 0:05; vs. baseline (Transfusion n = 7; Control n = 4).

Figure 3 Systemic inflammatory cytokine levels of TNF-alpha (A) and IL-6 (B).

Parameters are displayed as pg/mL. The systemic inflammatory cytokine levels did not differ between both study groups. There was a significant decrease in TNF-alpha expression compare to baseline values in both groups; * =p < 0:05; vs. baseline (Transfusion n = 7; Control N = 4).

Figure 4 Regional distribution of tidal volumes.

Non-dependent (A), central (B) and dependent (C) lung areas (% of the global tidal amplitude). No significant differences (each p > 0:05). The regional distribution of tidal volumes did not differ between both study groups (Transfusion n = 7; Control n = 4).

Figure 5 Pulmonary mRNA expression of inflammatory mediators.

Cryopreserved lung samples were collected after the experiment from lower (A, C, E) and upper (B, D, F) lung sections. mRNA expression of inflammatory parameters (IL-6 (A, B), TNF-alpha (C, D), INOS (E, F)) were analyzed via real-time polymerase chain reaction. mRNA expression is normalized to peptidylprolyl isomerase A (PPIA); no significant intergroup differences (each p > 0:05) (Transfusion n = 7; Control n = 4).

Figure 6 Cerebral mRNA expression of inflammatory mediators.

Cryopreserved cerebral samples were collected after the experiment from cortex (A, C, E) and hippocampus (B, D, F) regions. mRNA expression of inflammatory parameters (IL-6 (A, B), TNF-alpha (C, D), INOS (E, F)) were analyzed via real-time polymerase chain reaction. mRNA expression is normalized to peptidylprolyl isomerase A (PPIA); No significant intergroup differences (each p > 0:05) (Transfusion n = 7; Control n = 4).

Discussion

The present study establishes a novel model for characterization of remote organ injury related to the distribution of inflammatory mediators through the systemic circulation. We found that one-time high-volume allogeneic blood transfusion in healthy pigs neither impairs pulmonary integrity nor mediates inflammatory response within the lung, brain, or systemic circulation. Accordingly, random allogeneic blood transfusion in pigs is feasible, and per itself does not trigger inflammatory response in the acceptor animals. Hence, this model could be an appropriate approach to identify the main mediators in injurious organ interactions, i.e., by transfusion of blood from severely brain or lung injured subjects to healthy ones without cofounders. Other experimental protocols with comparable porcine transfusion models investigate questions like acute kidney injury or effects of blood storing and cooling. Several similar but, in some points, different protocols have been described that make use of autologous instead homologous blood, heavier animals to minimize the need for donor animals, or previously induce a hemorrhagic shock to reduce the rate of circulatory overload (Biagini et al., 2017; Masuda et al., 2018; Patel et al., 2011; Wozniak et al., 2018).

In principle, blood transfusion can be associated with severe side effects. Aside from incompatibilities, particularly pulmonary complications like transfusion-related acute lung injury (TRALI) or transfusion-associated circulatory overload (TACO) are described. Other systemic side effects include hypotension, anaphylaxis, hemolytic reactions and transfusion-transmitted bacterial infections (Bux & Sachs, 2008). Our study reveals a significant increase of mean arterial, pulmonary arterial and central venous pressure immediately following transfusion, which was underlined by increase of hemoglobin and hematocrit. These symptoms comply with subclinical TACO induced by supraphysiological increase of intravascular volume (Roubinian et al., 2017). Transfusion of longer stored but not freshly drawn blood was shown to considerably increase pulmonary arterial pressure which is presumably an effect of mediators released through storage (Baron et al., 2012; Solomon et al., 2013). In our study, similar but short-termed effects occurred despite blood transfusion immediately after withdrawal. Furthermore, fluid overload, which inevitably in our study, is also associated with pulmonary arterial hypertension (Yilmaz et al., 2016; Wolsk et al., 2019). As we found no coincident pathologies that allow distinguishing between fluid overload and mediator-based effects, identification of the exact mechanism is beyond the scope of this study.

Relevant impairment of the pulmonary gas exchange or alterations in extravascular lung water content, functional residual capacity or ventilation distribution were not observed after blood transfusion. Over eight hours PaO2/FiO2 decrease identifications to a certain degree alongside a raised alveolar-arterial oxygen difference. In the absence of histopathologic lung injury and during low-grade invasive mechanical ventilation, this is most likely a consequence of atelectasis formation that inevitably occurs during general anesthesia, muscular paralysis and mechanical ventilation (Hedenstierna & Edmark, 2015). Two animals had a slightly reduced PaO2/FiO2 ratio (<400 mmHg) already during baseline measurements without any signs for pulmonary or systemic illness: these two animals did neither respond differently to the transfusion nor showed signs of increased histopathologic lung damage despite formally exhibiting a PaO2/FiO2 <300 mmHg after eight hours. Increase and withdrawal of intravascular fluid volume in pigs was shown to alter the endexpiratory lung volume as measured by EIT, but did not affect tidal volume or ventilation distribution, which is in line with prior results (Bodenstein et al., 2012). TRALI represents another facet that directly affects the lung following blood transfusion. It is characterized by respiratory distress, hypoxemia, and non-cardiogenic lung edema that is induced by leukocyte-antibodies and neutrophil-priming substances (Bux & Sachs, 2008). These interact with neutrophil granulocytes, which thereby are activated. The resulting clots impair pulmonary capillary perfusion and induce capillary leakage by releasing toxic enzymes and reactive oxygen species (Sachs et al., 2006; Toy et al., 2005). We did not find any TRALI surrogates such as edema formation or increasing counts of leucocytes or specifically neutrophil granulocytes in our porcine model. Furthermore, the histopathological analysis did not reveal relevant inflammatory cell immigration. Additionally, the regional tidal volume distribution was unaltered, which is described as sign of severe pulmonary illnesses (Chiumello et al., 2013). Gender-specific forms of TRALI are particularly relevant in multiparous women as blood donors (Palfi et al., 2001; Schmickl et al., 2015). Hence, the exclusive use of male pigs may represent an approach to minimize the TRALI risk. TACO on the other hand tends to appear primarily in patients suffering from serious pre-existing illnesses. In our study healthy animals with intact pulmonary vascular barrier and competent kidney function were used, which enables rapid compensation of the intravascular overload (Bux & Sachs, 2008; Roubinian et al., 2017).

Single organ injury or failure can subsequently affect the integrity of distant remote organs leading to multiple organ failure. In particular, deleterious interactions of the lung and the brain have been described (Pelosi & Rocco, 2011). Several models suggest that the lung may play a pivotal role as causative but also remotely injured organ: in mechanically ventilated rats that underwent an aggressive ventilator regime showed remarkable cerebral inflammatory response (Quilez et al., 2011). Cyclic variations of the PaO2 caused by non-protective ventilation in lung injured pigs are transmitted to the cerebral tissue and may trigger cerebral injury (Klein et al., 2013; Klein et al., 2016). Furthermore, also a cerebral insult is capable to induce or aggravate lung injury (Heuer et al., 2011). Even mechanical ventilation by itself can represent a trigger for cerebral cytokine expression (Kamuf et al., 2018). Accordingly, we also assessed brain inflammatory response measured by cortical and hippocampal mRNA expression of key inflammatory mediators, which also were unaffected by the transfusion procedure. In comparison to human blood the use of porcine model may offer several advantages for studies concerning remote organ injury: Unlike human blood, where transfusion of incompatible blood products is deleterious, pig blood lacks the human blood group B, the secondary absorption of antigens from the plasma to the blood cells and the Rh polymorphism (Omi et al., 2003). Additionally human blood is polygene coded, which results in a much higher incompatibility potential (Smith et al., 2006).

The study has several limitations. The study was not designed to assess long-term effects of blood transfusion but to exclude acute side effects of allogeneic blood transfusion itself with regard to a novel experimental approach to elucidate pathomechanisms of deleterious remote organ injury. Logistically it was not possible to investigate different pig subspecies or breeds. Hence, generalization of the reported findings lies beyond the scope of this study. Animal care reasons and the need for additional blood donor animals required a strict limitation of the animal numbers, whereas the chosen group numbers were in line with previous studies of inflammatory acute effects in pigs (Hartmann et al., 2015; Ziebart et al., 2014). Exchange of the transfused blood by simultaneous blood withdrawal may be a possibility to avoid fluid overloading. However, given the risk of unpredictable removal of the donor blood, hemodynamic variations, and potential microcirculatory or inflammatory impairment during this complex procedure, we decided to decline this option. Small volume withdrawal prior to the transfusion procedure may represent a reasonable compromise, and may reduce the risk of TACO (Masuda et al., 2018). The use of plasma instead of whole blood is an interesting alternative approach to optimize this setting: theoretically, plasma administration will not increase the hemoglobin content, whereas full concentration of potentially inflammatory mediators should be present. This approach, however, was not feasible for the present study. Additionally, we did not measure cytokine contents in the transfused blood, as the donor animals were healthy and the blood was transfused within minutes. These measurements need to be included in studies focusing on severely injured donor animals.

Conclusion

In conclusion, high volume random allogeneic blood transfusion in pigs neither impaired pulmonary integrity nor induced systemic, lung, or brain inflammatory response. No further side effects beyond expectable and temporary alterations of hemodynamics and blood count related to the rapid increase of blood volume were detected. Random allogeneic blood transfusion in pigs may represent an experimental model to characterize the blood-bound transmission in remote organ injury.

Supplemental Information

Dataset S1 Histopathological parameters

Click here for additional data file.

Dataset S2 blood gas parameters

Click here for additional data file.

Dataset S3 hemodynamic parameters

Click here for additional data file.

Dataset S4 spirometric parameters

Click here for additional data file.

Dataset 55 biochemical parameters

Click here for additional data file.

The authors thank Dagmar Dirvonskis and Dana Pieter for excellent technical support. Parts of this study are included in the doctoral thesis of Moritz Schaefer, and content of the professorial dissertation (habilitation) of Alexander Ziebart. Preliminary results of the study were presented in poster sessions at the German Anesthesia Congress (DAC), Nuernberg, Germany in 2017.

Additional Information and Declarations

Competing Interests

Author Contributions

Animal Ethics

Data Availability

The authors declare there are no competing interests.

Alexander Ziebart conceived and designed the experiments, performed the experiments, analyzed the data, contributed reagents/materials/analysis tools, prepared figures and/or tables, authored or reviewed drafts of the paper, approved the final draft.

Moritz M. Schaefer performed the experiments, analyzed the data, contributed reagents/materials/analysis tools, prepared figures and/or tables.

Rainer Thomas, Jens Kamuf, Andreas Garcia-Bardon and Christian Möllmann performed the experiments.

Robert Ruemmler prepared figures and/or tables, approved the final draft.

Florian Heid authored or reviewed drafts of the paper, approved the final draft.

Arno Schad analyzed the data.

Erik K. Hartmann conceived and designed the experiments, contributed reagents/materials/analysis tools, prepared figures and/or tables, authored or reviewed drafts of the paper, approved the final draft.

The following information was supplied relating to ethical approvals (i.e., approving body and any reference numbers):

The prospective-randomized animal study was approved by the State and Institutional Animal Care Committee (Landesuntersuchungsamt Rheinland-Pfalz, Koblenz, Germany) (approval number: 23 177-07/G 14-1-084).

The following information was supplied regarding data availability:

The raw data are provided as Supplemental Files.

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
