# Peer review of "Random allogeneic blood transfusion in pigs: characterisation of a novel experimental model"

_PeerJ, doi:10.7717/peerj.7439_

## Round 0.1 · original submission · Major Revisions

Your study is of interest, but as you can see, the reviewers had extensive suggestions.

I hope you can revise the manuscript to address their comments

Reviewer 1 ·

Basic reporting

1. In general, the clear, unambiguous, professional English language was used throughout; however:
a. It could at times be more professional/scientific. Examples include lines 72-73 (“… and therefore hardly represents a relevant blood group antigen”; line 155 “… no hints for systemic inflammatory response were found…”; Lines 194 – 196: “Hints for … are missing.”;
b. Confusing terminology was sometimes used. e.g. “Whole blood” should be used rather than “full blood” in several instances.
c. Numbers at the start of sentences should be written rather than as numerals. E.g. line 125: “Sixteen…” rather than “16…”
d. The sentence across lines 84 to 86 is confusing as it implies that the guidelines apply to the dates 12/2014 till 3/2015, whereas the authors presumably mean that the experiments that they conducted were performed between these dates.
e. The sentence across lines 170 and 171 “The tissue wet to dry ratio as surrogate of edema formation was not increased following transfusion” is confusing as wet to dry weights were not performed pre and post-transfusion, rather they were performed on post-mortem tissue across different groups of pigs.
f. What is the context for the word “single” in line 179 within the sentence “We found that single high-volume allogeneic blood transfusion in healthy pigs…”?
g. While PeerJ does not specify it in their author instructions, I would suggest that an in-text reference at the end of a sentence should be placed before rather than after the full stop. This better links the reference to the statement that it applies to.

2. Other studies have reported on porcine transfusion models (e.g. Patel et al. 2011. Am J Physiol Renal Physiol; Biagini et al. 2018. Braz J Med Biol Res; Wozniak et al. 2018. Anesthesiology; and Masuda et al. 2018. Vox Sang.), albeit not in the context of organ cross-talk. These previous models should be described in the introduction.

3. The structure of the manuscript was logical and facilitated comprehension of the study and its findings.

4. In general the figures and table were well-designed and relevant; however, figure legends were insufficient for the reader to interpret the figures without reference to the manuscript text. Please address this limitation. Please also see points below for where clarifications and improvements are suggested:
a. Figure 2: Is this missing a 0h time-point? Are the decreases in haemoglobin and haematocrit in the control group significant? Do the asterisks represent P<0.05 vs. baseline or vs. control group?
b. Figure 3: Is this missing a 0h time-point? Do the asterisks represent P<0.05 vs. baseline or vs. control group?
c. Figure 4: should the y-axis title say “relative impedance” rather than “realtiv impedance”? Would it be more appropriate to have a graph for each lung area so that transfusion vs. control can be compared?
d. Figure 5 and Figure 6: There is a lot or unnecessary repetition of labels in these figures. For example in Figure 5: “lower lung” and “upper lung” could just be stated once above the appropriate column and no need to state the analyte for each graph title; “IL-6”, “TNF-alpha” and “iNOS” could just be stated once per row, and again no need to have this as a graph title; “control” and “transfusion” could also just be stated once per column rather than on every graph.
e. Table 1: This is a very large table. It may be an idea to split into two tables – maybe one for haemodynamic data and one for respiratory data.
f. Table 2 and Figure 2 appear to present the same data, just that in the table it is the actual results, while in the figure it is transformed to % baseline. Is this correct? If so are both necessary?

5. Raw data is included as supplements. Some additional labelling would facilitate interpretation of these data.

Experimental design

1. The manuscript meets the scope of PeerJ.

2. The research question ultimately posed by the authors is whether blood-borne factors can precipitate organ cross-talk. More specifically, the research question posed by the authors in the manuscript is whether a porcine model of whole blood transfusion is a suitable model in which to investigate the mechanisms of organ cross-talk.

3. In general the study described by Ziebart et al. meets acceptable technical and ethical standards; however, I do have some concerns that need clarification:
a. It appears that the transfusion was of leucodepleted whole blood. Yet inflammatory cytokines are hypothesised to be responsible for initiating organ cross-talk via the blood stream. Did the authors consider processing the whole blood into plasma which could be frozen for future experiments? This approach would have allowed increased flexibility in the time intervals between whole blood collection and the transfusion experiments. And the inflammatory cytokines would still be present in the plasma. Furthermore, this approach would have avoided the increased in Hb and Hct evident following whole blood transfusion.
b. Other animal models of blood transfusion utilise blood collection from conscious donor animals without the need for euthanasia, and this seems to be a more ethical approach. Why were the donor pigs euthanised following blood collection? Was this due to the volume of blood collected, and if so what percentage of estimated circulating blood volume did this represent? If too high, could this have been avoided by using older pigs as the donors (I understand that adult German Landrace pigs reach 200 – 300kg in weight). For example, in Australia and the UK, the minimum weight requirement for blood donors is 50 kg. Assuming a circulating blood volume of 70 mL/kg, then collection of a 450 mL unit of whole blood represents 13% of the circulating blood volume. Furthermore, if the experiment used plasma and not whole blood (see earlier point), then units could be collected over time for subsequent experiments, reducing the number of donor pigs required and possibly making scheduling experiments more flexible.
c. In the discussion (lines 238 – 242) the authors state that they did not perform “simultaneous blood withdrawal” to avoid risk of circulatory overload as this would also have removed factors within the transfused blood, and because of the haemodynamic changes and complexity of the procedure. Would withdrawal of a smaller volume of blood prior to transfusion been an alternative to reduce risk of circulatory overload while at the same time minimising these other risks?

4. In general an appropriate level of detail in the Materials and Methods section was provided; however, some aspects do require clarification in the manuscript.
a. Blood collection: should state the average (+/- SD) volume of blood collected. What percentage of estimated circulating blood volume does this represent? If not a multiple of 450 mL then how did this impact the ratio of whole blood blood to anti-coagulant? Is it scientifically valid to transfuse whole blood from ‘underweight’ collections?
b. It is stated that whole blood was collected into Composelect blood packs, but these come in two formats – one in which the whole blood is leucofiltered prior to processing, and one in which the whole blood is separated and the red cells are leucofiltered. I assume that the former were sued; however, the manuscript should clearly state which ones were used. If the former, was the whole blood leucofiltered prior to storage at 4C? If the latter, then how was the whole blood separated? Furthermore, it should be stated clearly in the manuscript that these blood packs are CPD anticoagulated.
c. How long was the donor blood stored prior to transfusion? Were cytokine levels measured in the blood to be transfused?
d. How was the transfusion performed? Was it warmed prior to transfusion? Was it administered via a transfusion giving set (with a filter to remove aggregates)? What was the average volume (+/- SD) transfused?
e. How long was the baseline period? Was this sufficient time for the animals to recover from the instrumentation?
f. What type of blood samples were collected throughout the experiment? E.g. were they EDTA, citrate or heparin?
g. The reference gene used in the real-time PCT experiments should be specified in the Materials and Methods section and not just in the Figure 6 legend

Validity of the findings

1. Ziebert et al. present a novel and interesting study describing the esablishement of a porcine model in which they plan to investigate the hyposthesis that blood-borne factors can mediate organ cross-talk. Presumably they will do so by transfusing whole blood from an injured pig into a healthy pig and observing the transfused pig for the development of injury. In the data presented, transfusion of whole blood between healthy pigs proceeded without evidence of acute transfusion reactions or haemodynamic, inflammatory or respiratory complications.

2. In general the data presented were robust, statistically sound and controlled. However the following points require clarification.
a. The methodology states that FiO2 was set to 0.4 (line 100) yet in table 1 of results FiO2 is reported throughout to be 0.39. Why the discrepancy? Furthermore, what was there any protocol to alter ventilation settings in line with respiratory function?
b. Was physiological data collected more frequently than the four time-points for which results are presented (baseline, t=0, t=4 and t=8)? The sentence in the discussion (lines 190 – 191) suggests that in addition to the increase in mPAP, transient increases in MAP and CVP were observed; however, these are not evident in the data presented in Table 1, nor in the raw data included as supplement 1. I would suggest that if more frequent data (e.g. hourly) are available that these be included in the manuscript.
c. As per the previous point, it appears that arterial blood gas samples were collected and analysed more frequently than presented in Table 1. Again, I would suggest that including these data rather than presumably averaging out measurements over a number of arterial blood gas samples would be preferable, even if only as supplementary data.

3. Where the authors have extrapolated from the findings of their study, the speculations they have made appear for the most part to be reasonable. However, some statements supporting their extrapolations require clarification.
a. The authors state that “Appearance of TRALI depends on the gender of the blood donor, and is particularly relevant in multiparous women.” However, TRALI may be either antibody-mediated, in which case the statement is true, or it may be mediated by biological response modifiers related to blood component storage duration. Given the latter mechanism, it is important for the methodology to be clear on the length of tiem that the blood was stored at 4C prior to transfusion. Furthermore, measurement of cytokine levels in the blood to be transfused would have further minimised the risk of TRALI in the porcine model.
b. The authors state that “Animal care reasons and the need for additional blood donor animal required a strict limitation of the animal numbers…” however, I would argue that it is unethical to limit the number of subjects per group if it means that a study is under-powered to answer its primary question. As per the earlier comment, I think that the blood donor part of the study could have been better designed from both an ethical and a technical perspective.

4. The conclusions of the study are reasonably well stated; however, the following
a. Statements such as “high volume… blood transfusion” are difficult to make in the context of the manuscript not stating either the normal circulating blood volume of German Landrace pigs or the percentage of blood transfused.

Additional comments

The manuscript submitted by Ziebart at al. describes the design of a porcine model suitable for the subsequent investigation of organ cross-talk. Presumably they will do so by transfusing whole blood from an injured pig into a healthy pig and observing the transfused pig for the development of injury. In the present study, transfusion between healthy pigs proceeds without evidence of unexpected haemodynamic, inflammatory or respiratory complications, with the caveat that a transient pulmonary arterial hypertension was observed.

Some general points to address:
1. Typographical errors: line 72: “gene” not “gen”; line 74: “barely” not “barley”; line 86: “Landrace” not “land race”; presumably the word “blood” is missing in the sentence from lines 238 – 239;
2. Numbers at start of a sentence should be written: line 86: “Twenty” not “20”;
3. Line 86: “Sus scrofa domestica” should be italicised
4. Whether “granulocyte clots” form as part of the pathogenesis of TRALI is not clear. Please amend the sentence from line 202 – 204.

Reviewer 2 ·

Basic reporting

• Clear and unambiguous, professional English is used throughout the article
• Literature references are sufficient; sufficient information about the field background.
• The article includes an introduction and background to demonstrate how the work fits into the broader field of organ crosstalk. The relevant prior literature is appropriately referenced.
• From the reviewers point of view the phenomenon of brain-lung interaction in men and pigs needs to be introduced a more detail, including potential mechanisms and risk factors. Some typical clinical presentations in the ICU setting would probably be helpful.
• The hypothesis and results are discussed as a “plus theory”, based on the notion that transfusion represents the systemic exposure to an “injectable” active component responsible for the cross-talk. However, a “minus theory” is possible, where transfusion impair immunocompetence and confers immunosuppression responsible for various observations during organ cross-talk. Even if transfusion is accompanied with proinflammatory signs in humans, the examination of the activity of specialized T cells could be discussed.
• The way the main hypothesis is described is not sound and remains unclear. One pathophysiological mechanisms of organ cross-talk includes the release of mediators from an injured organ into the blood stream. However, it remains unclear why allogenic blood transfusion would mimic the mechanisms of organ cross-talk because blood transfusion does not necessarily imply that active compounds (mediators) are being transfused. Even if transfusion was associated with systemic infusion of active compounds, it is not clear whether these compounds represent the mediators that are released by organ A to induce remote organ injury in organs B, C, or D. Therefore, the main objective of the present study is limited to the characterization of the baseline immunogenic effect of allogeneic blood transfusion in pigs (line 75 ff.) It is a transfusion study. Not an organ cross-talk study, and therefore, the statement (line 177) that this study established a novel model for the characterization of remote organ injuries is not valid and should be removed. Furthermore, it is suprising and not substantiated in the introduction why the authors hypothesize that transfusion “neither inhibits inflammatory response nor impairs the pulmonary function” (line 77ff). The scientific rationale why transfusion would impair pulmonary function needs to be mentioned in the introduction.
• Professional article structure, figs, tables. Raw data shared is given.
• The structure of the article is in agreement with the format of ‘standard sections’
Figures are relevant to the content of the article, of sufficient resolution, and appropriately described and labeled.
All appropriate raw data have been made available in accordance with the Data Sharing policy.
• The submission is ‘self-contained,’ and represents an appropriate ‘unit of publication’, and includes the results relevant to the hypothesis.
• Line 54: “well-know” and “not fully understood” are potentially contradictory
• Line 233: again, the interpretation that this is a study about organ interaction should be removed because the design of the study does not include organ interactions
• Line 249: I agree with the authors that blood transfusion “may represent a (novel?) model to characterize remote organ injury”, unfortunately, this study does not help answer this hypothesis. Future studies are needed to address this hypothesis, but the aspect or organ cross-talk should be removed from this manuscript.
• Figure 2: absolute leukocyte counts and hemoglobin level should be given
• Figures 5 and 6: The normalized mRNA expression should be displayed in relation to controls (=1), otherwise it is very hard to interprete differente scaling of the various graphs. The differentiation between lower and upper lung lobes and cortex vs. hippocampus are not discussed. Why is it important to make that differentiation and how can the results be interpreted?

Experimental design

• Original primary research within Aims and Scope of the journal.
• The submission has defined the research question, the relevance remains slightly unclear because the notion of organ cross-talk is not defined specifically
• The conclusions include the hypothesis that remains unanswered
• The reviewer feels that the study time of 8h between transfusion and end of study needs to be explained in the context of the brain Lung crosstalk and the cited lterature. Why did the authors choose this interval? In the cited paper: Systemic Pao2 Oscillations Cause Mild Brain Injury in a Pig Model by Klaus U. Klein et al. the authors found only a mild cerebral inflammation after a 20 h protocol of pigs treated with ECMO, wich itself is a device with a relevant inflammatory impact. What is the translational relevance of the present hypothesis and study design?

• The investigation is performed to a high technical & ethical standard.
• The research has been conducted in conformity with the prevailing ethical standards in the field.
• Methods described with sufficient detail & information to replicate.
• Only the procedure of drawing donor blood needs smoe additional details. The authors used a leucocyte depleting collector bag and stored the bags at 4°C. Hence the blood transfusion is a composite of red cells and platelets. The authors should render this more precisely and add the duration of blood storing or the timeline. Did they start on day 1 -4 with blood donation followed by day 5 – 10 with transfusion or did they perform blood donation and transfusion on the same day. This information will help to reproduce the study model by another investigator.
• Why did the authors before “top-load” blood transfusion rather than transfusion after withdrawal of an equivalent amount of blood. Top-load transfusion may dilute the effect of potential mediators that are being transfused and may cause inflammatory organ injuries. Top-load infusion of electrolyte solution creates lesser dilution of baseline circulating mediators as compared to electrolyte infusion after withdrawal of an equivalent amount of blood volume. The study design of transfusion and electrolyte infusions is therefore reducing the probability of significant differenced between both groups with respect to circulation inflammatory parameters and functional assessment of remote organs (lung, brain).

Validity of the findings

See above re hypothesis and rationale, and limited validity of last sentence of conclusions.

Additional comments

See above.
I suggest to rewrite the article, giving it a different perspective and stressing the assessment of immonogeneic effects of transfusion, NOT of organ cross-talk, because the latter is misleading and not supported by the design and the results of the study.

---

## Round 0.2 · Minor Revisions

Thank you for your submission and revision. Your manuscript is acceptable in provision. However, a small discrepancy as raised by a review should be addressed. Therefore,I am returning the revision for your careful response.

Thank you
Sincerley
Cheorl-Ho Kim
A dealing editor

Reviewer 1 ·

Basic reporting

1. In general, the clear, unambiguous, professional English language was used throughout; however:
a. It could at times be more professional/scientific. Examples Lines 40 – 41: “No hints for systemic inflammatory response....”
b. When data values are written, ‘3.5’ rather than ‘3,5’ should be used to describe 3 and a half as this is the convention in English.
c. Some examples of odd wording. Example line 125 “piped”

2. The Introduction places the study into the context of organ cross-talk. Literature was well referenced & relevant.

3. The structure of the manuscript was logical and facilitated comprehension of the study and its findings.

4. In general the figures and tables were well-designed and relevant; however, figure legends were insufficient for the reader to interpret the figures without reference to the manuscript text. Please address this limitation. Please also see points below for where specific clarifications and improvements are suggested:
a. Figure 1: The figure legend requires some clarifications. The duration of the baseline period should be noted in the figure (from the manuscript this appears to be 20 minutes). Abbreviations (e.g. EIT) should be defined in each figure even if they have already been defined in the manuscript. In the sentence “The hemodynamic and spirometric parameters were recorded permanently.” is ‘permanently’ the correct word? Perhaps the authors mean ‘continuously’? The sentence “Lung and brain were analysed histopathological and biochemical after the end of the experiment” appears incomplete. Furthermore, what is meant by “biochemical” as no reference is made to biochemical analyses in the manuscript’s Materials and Methods section.
b. Figure 2: If blood samples were collected every 2 hours and underwent hematological analysis, then all of these data should be shown on the graphs. In particular, the 0h time-point that occurs immediately following transfusion should be included. An immediate increase in hemoglobin, hematocrit and erythrocytes would be expected with the whole blood transfusion. The description of what the asterisks represent should come at the end of the figure legend rather than at the start.
c. Figure 3: If blood samples were collected every 2 hours and underwent ELISA analysis, then all of these data should be shown on the graphs. The description of what the asterisks represent should come at the end of the figure legend rather than at the start. Do the asterisks represent P<0.05 vs. baseline or vs. control group, presumably the latter? The figure legend states that “Parameters are displayed as percent of baseline” however the data in the graphs are presented as pg/mL. It is unclear which group the asterisks in the TNF-alpha graph refer to: the control group or the transfusion group? Are the symbols in the IL-6 graph correct as it appears that black circles are used for one of the data-sets rather than grey circles as indicated in the figure legend.
d. Figure 4: If the comparison is being made between control and transfusion groups then there should be a graph for each lung area so that transfusion vs. control can be on the same graph.
e. Figure 5: The second and third sentences of the figure legend are repetitive of each other. The figure legend should specify that these analyses were based on cryopreserved lung samples, as well as very briefly describe the procedures used to generate these data.
f. Figure 6: As per Figure 5.
g. Table 1: Please refer to earlier comments on the presentation of values such as “3,6” and the inclusion of only selected time-points. The table legend states that “No significant intra- and intergroup changes of hemodynamic parameters were observed after blood transfusion or fluid infusion, apart from temporary increase of the pulmonary arterial pressure after transfusion.” however, CVP was also increased after transfusion compared to baseline but without being increased compared to the control group.
h. Table 2: Please refer to earlier comments about the presentation of values such as “6,2” and the inclusion of only selected time-points.
i. Table 3: Please refer to earlier comments about the presentation of values such as “7,5” and the inclusion of only selected time-points. The two sentences of the figure legend are contradictory. Furthermore, while lactate levels decrease, AaDO2 levels actually increase. Values for pH should be provided in more detail, i.e. 7.456 rather than 7.5.

5. Raw data is included as supplements. Some additional labelling is still required to facilitate interpretation of these data. For example, in DataSetS1 presumably each worksheet represents a different experiment; however it is unclear to what the numbers “1 – 4” represent. Presumably dependent periphery, dependent central, non-dependent central, and non-dependent periphery; however which is which? What do the four different values represent – four different fields of view? The summary worksheet should specify which experiment the values come from. Grouping should be also be displayed. TableS1, S2 and S3 are datasets rather than tables?

Experimental design

1. The manuscript meets the scope of PeerJ.

2. The research question ultimately posed by the authors is whether blood-borne factors can precipitate organ cross-talk. More specifically, the research question posed by the authors in the manuscript is whether a porcine model of whole blood transfusion is a suitable model in which to investigate the mechanisms of organ cross-talk.

3. In general the study described by Ziebart et al. meets acceptable technical and ethical standards; however, I do have some concerns that need clarification:
a. Data were collected more frequently than presented in the figures/tables. For example, arterial blood gases were collected hourly (as per TableS1) however, only Baseline, 0h, 4h and 8h data were included in Table 3, meaning that 6 data points were excluded. What is the rationale for collecting these data but not presenting or analysing them? Would including these data alter the findings of the study?
b. Based upon total number of pigs (20), number of donor pigs (8), and number of control pigs infused with fluid (4) it can be inferred that there were eight pigs transfused with whole blood. This isn’t clearly stated anywhere in the manuscript. Reviewing the supplementary data and tables it appears that lung sections were scored for 12 pigs (presumably 4 controls plus 8 transfused), but tables S1, S2 and S3 don’t have data for experiment 3. This leaves the transfusion group with only 7 pigs. Please clarify these numbers in the manuscript, and if data from experiment 3 was not included in the manuscript then the rationale for its exclusion should be provided, and data from this experiment should also be removed from the histological results.

4. In general an appropriate level of detail in the Materials and Methods section was provided; however, some aspects do require clarification in the manuscript.
a. Blood withdrawal: Isn’t the CPD anticoagulant present in the primary pack (into which the blood is collected) and the SAGM additive solution in the satellite pack (into which the leucoreduced whole blood is passed)? Furthermore, should the leucocyte reduction filter have been pre-wet by passing the SAGM through into the primary pack before passing the whole blood/CPD/SAGM mixture back through the filter into the satellite pack?
b. Experimental protocol: Information presented in this section may be better placed in the Anesthesia and Instrumentation section. For example the information from lines 140 – 151 regarding monitoring and background Sterofundin infusion.
c. Post mortem analysis: Was it serum or plasma? Both are stated in line 157.

Validity of the findings

1. Ziebert et al. present a novel and interesting study describing the establishment of a porcine model in which they plan to investigate the hypothesis that blood-borne factors can mediate organ cross-talk. Presumably they will do so by transfusing whole blood from an injured pig into a healthy pig and observing the transfused pig for the development of injury. In the data presented, transfusion of whole blood between healthy pigs proceeded without evidence of acute transfusion reactions or haemodynamic, inflammatory or respiratory complications. However, according to raw data presented in TableS1, two of the transfused animals developed hypoxaemia (PaO2/FiO2 < 300) following transfusion, although these were the only two animals that had a baseline PaO2/FiO2 of <400. This should be presented in the results and discussed in the discussion. Did either of these two animals also display evidence of pulmonary edema (by histology or wet/dry weight anlaysis)? If so then transfusion-related acute lung injury would be indicated.

2. In general the data presented were robust, statistically sound and controlled. However the following points require clarification.
a. In my experience all of the time-points for which data are collected are presented in a manuscript. However, in the figures and tables only data from specific time-points are presented in this manuscript. While these additional data points may not increase the power of the study nor provide new findings, these data are important to provide a more complete picture of the experiments and for the reader to be satisfied that the conclusions made by the authors are valid.
b. The sentence in the Discussion (lines 218 – 220) suggests that in addition to the increase in mPAP and CVP following transfusion, an increase in MAP was also observed; however, this is not evident in the data presented in Table 1.

3. Where the authors have extrapolated from the findings of their study, the speculations they have made appear for the most part to be reasonable. However, some statements supporting their extrapolations require clarification.
a. Further to the earlier point, what are the implications in that 2 out of the 7 or 8 pigs in the transfusion arm developed hypoxaemia?
b. The authors state that “Appearance of TRALI depends on the gender of the blood donor, and is particularly relevant in multiparous women.” However, TRALI may be either antibody-mediated, in which case the statement is true, or it may be mediated by biological response modifiers related to blood component storage duration in which case no differences between female and male donors have been reported.
c. What are the implications of the transient increase in mean pulmonary arterial pressure? This has previously been associated with the transfusion of soluble factors present in stored but not fresh red cell concentrates (Tung et al. Critical Care Medicine 2012; Baron et al. Anesthesiology 2012; Solomon et al. Blood 2013; Fung et al. Vox Sanguinis 2013) and yet the whole blood transfused into the pigs was fresh.

4. The conclusions of the study are reasonably well stated; however, the implications of the hypoxaemia development need to be considered prior to being able to make statements such as “high volume random allogeneic blood transfusion in pigs neither impaired pulmonary integrity…”.

Additional comments

The manuscript submitted by Ziebart at al. describes the design of a porcine model suitable for the subsequent investigation of organ cross-talk. Presumably they will do so by transfusing whole blood from an injured pig into a healthy pig and observing the transfused pig for the development of injury. In the present study, transfusion between healthy pigs was reported to proceed without evidence of unexpected haemodynamic, inflammatory or respiratory complications; however, a transient pulmonary arterial hypertension was observed and two of the transfused pigs developed hypoxemia (PaO2/FiO2 < 300) post-transfusion.

Reviewer 2 ·

Basic reporting

I gladly accept the opportunity to re-review the revised manuscript of Ziebart et al. entitled “Random allogeneic blood transfusion in pigs: characterisation of a novel experimental model”.
Based on the authors’ revision, I have carefully reviewed the tracked-changes version of the manuscript as well as the point-by-point answers to the reviewer’s previous comments. I would like to thank the authors for the detailed answers. While the answers are straight-forward and clear, and the manuscript has been changed and adapted mainly in the methods and results section, I regret to note that the authors have not taken the opportunity to clarify one very crucial and misleading point of their manuscript. And I had included this comment in my previous review:
The crucial point is that the authors study, its design and results, are valid to assess the immunogeneic effects of whole blood transfusion. Therefore, the title is correct by stating that this is description of a (new) animal model. However, the authors’ study is NOT a study investigating organ cross-talk, not at all. And therefore, any mention of organ cross-talk remains purely speculative, is not supported by their results, and should be removed from the manuscript. One single sentence at the end of the discussion, i.e. that immunogeneic effects of blood transfusion may play a role in designing adequate models for the study of organ cross-talk mechanisms during transfusion, is a valid point, and is just enough to make the point.
In their point-by-point reply, the authors argue that upcoming studies (on organ cross-talk) would make use of this model. I agree, that this may be an option for future studies, and that these upcoming studies rely on the basic immunogeneic effect of whole blood transfusion as a “control”. In this context, the present results may represent a basic definition and description of a model, that – in the future - may be used as a “control” to study whether organ cross-talk may be mediated by whole blood transfusions. Based on the considerations, again, I suggest to rewrite the article, giving it a different perspective and stressing the assessment of immonogeneic effects of transfusion, NOT of organ cross-talk, because the latter is misleading and not supported by the design and the results of the study. Preferably, however, I advised the authors to put two things together in one comprehensive manuscript: 1. the “baseline characteriziation” of the model presented in the present manuscript and 2. their results of upcoming studies investigating whether blood transfusion is involved in the mechanisms of organ cross-talk. I have gotten the impression that the current definition of their (new) transfusion model is just the baseline, and indispensable characterization of a model, that shall be used to study the initial hypothesis. Therefore, these two aspects belong in one manuscript.
Since this is a very basic and major concern I have about the current manuscript, please allow me to refrain from making additional specific comments. In general, there are a few typos; grammar, language should be revised, and comparisons in the results section should be made clearer. Moreover, the discussion should focus on the immunogeneic effects rather than on speculations about organ interactions. Figures 5 and 6 may be clearer, when normalized mRNA expression are being shown relative to controls (controls=1). Sample sizes should be added to tables and figures. Again, I strongly suggest to invest more energy to rewrite the manuscript to make a valid point and transfer a meaningful and clear message to the readers.

Experimental design

no additional comment

Validity of the findings

no additional comment

Additional comments

no additional comment

---

## Round 0.3 · accepted · Accept

Your manuscript will be accepted. Congraturations !!